# Imagining Decent Work towards a Green Future in a Former Forest Village of the City of Istanbul

İklil Selçuk [1], Zeynep Delen Nircan [2] and Burcu Selcen Coşkun [3,*]

1    Department of Humanities and Social Sciences, Özyeğin University, 34794 Istanbul, Turkey
2    Foundation Development Directorate, Sabanci University, 34956 Istanbul, Turkey
3    Department of Architecture, Mimar Sinan Fine Arts University, 34427 Istanbul, Turkey
*    Correspondence: selcen.coskun@msgsu.edu.tr

**Abstract:** This paper addresses issues pertaining to the future of work and sustainability through the lens of a case study of ecological deterioration and how it destroys and creates green jobs in a forest village of Istanbul. As elsewhere in major urban centres of developing countries, the hyper-expansion of city regions due to authoritarian developmentalism fosters the state-led construction sector in Turkey. Growth-driven economic policies continue to have adverse effects on the environment, resulting in deforestation among an array of ecological damage. Based on a qualitative analysis of oral history interviews and observations informed by a larger interdisciplinary research project, we observe resilience in the forest village under scrutiny as certain types of work are abandoned, and new forms are created by adaptation to the ecological and social conditions. The perceptions of changing conditions by locals vary across existing ethnic, gender, and class hierarchies in the local community. Moreover, our findings indicate that the types of work available in the village prior to urban transformation were not all decent or green. In face of ongoing ecological deterioration in a (formerly) forest community, participatory micro-initiatives, and grassroots, utilizing local community projects emerge that nevertheless pursue a green and just transition. We focus on one such initiative, the Community Fungi platform, to demonstrate the possibility of working towards a collective imagination of a green future inspired by past but unforgotten sustainable communal practices, in the context of the forest village under scrutiny in this paper.

**Keywords:** deforestation; commons; decent work; urban transformation; gated communities; feminisation of labour; green transitions; grassroots micro initiatives

## 1. Introduction

The ongoing debate on a just and equitable reversal of the climate crisis crosses the industrial Global North and developing Global South divide. While green alternatives regarding the industrialised North led to the post-capitalistic de-growth idea, the South is viewed in the light of environmental justice struggles defending old styles of living alongside pains of growth. As scholars suggest increasing the dialogue between de-growth and environmental justice perspectives (Chomsky et al. 2020; Singh 2019), the North–South divide is also criticised for constituting an unjust base for existing capitalist growth paradigms at the cost of nature and for its gender bias, underlined by feminists (Littig 2017). Forms of work constitute an integral part of this debate (Dordmond et al. 2021; Kouri and Clarke 2014; Maclean et al. 2018; Poschen 2015; Stilwell 2021; Tănasie et al. 2022) as future possibilities are also discussed with respect to the alternatives of creating eco-efficient green work and critical approaches that suggest the decommodification of work (Bottazzi 2019).

This paper aims to contribute to the future of work and sustainability debates that inform this Special Issue through a case study of a forest village, Ömerli, in the periphery of metropolitan Istanbul. As already observed in different parts of the city, uncontrolled urban transformation driven by neoliberal policies and authoritarian developmentalism

results in various environmental drawbacks, including deforestation in this area. Focusing on the forest village under scrutiny, we suggest that adaptation to the new circumstances by abandoning old types of work and creating new ones in gendered and ethnic hierarchical forms reflects the resilience of the forest community. A comparison of extinct and surviving types of work based on our informants' narratives portrays a complex picture regarding whether these jobs are green or decent. Observing the ongoing ecological deterioration in a (formerly) forest community within the broader transformation that renders a green and just transition increasingly difficult, we argue that participatory micro-initiatives and grassroots festivities that bring scholars and locals together, may still offer one possibility for change.

We thus look in granular detail into one such micro-initiative, Community Fungi (Mahalle Mantarı in Turkish), developed in the forest village under scrutiny, to foster interactive critical thinking among locals and scholars on the meaning of "green transitions". Community Fungi aims at providing a platform for "expressions of a collective imagination" on how a just, eco-friendly future may look, considering past local communal practices. This paper is based on action research and fieldwork conducted in the village as a product of the Community Fungi initiative. The following part situates the topics and concepts employed in the present research within the literatures on global developmentalist economic policies, middle-class suburbanisation, deforestation, extinction of traditional ways of working, and grassroots initiatives of community action.

## 2. Literature Review

Authoritarian governments with visions to "revive national greatness" have been leading neoliberal developmentalist agendas, which has been argued to represent an authoritarian turn in global politics (Arsel et al. 2021). The literature on authoritarian developmentalism cuts across levels of income and development transcending the North–South divide. Studies on authoritarian developmentalist policies feature cases including, but not restricted to, Brazil (Saad-Filho and Boffo 2021), Egypt (Adly 2021), Turkey (Adaman and Akbulut 2021), the Philippines (Ramos 2021), Hungary (Scheiring 2021), India (Sinha 2021), and the USA (Kiely 2021). Adaman and Akbulut, among these case studies, underline the resilience of Erdoğan's regime in Turkey supported by neoliberal values of prosperity framed as growth. In this context, growth is a function of the construction sector. Regime-affiliated corporations within the construction sector are conspicuously entangled with mechanisms that reward them. Endorsing the construction sector necessitates sanctioning "heavy environmental costs", painting an interdependent picture of authoritarian populism and developmentalism (Adaman and Akbulut 2021). A significant outcome of urban transformation driven by these populist—developmentalist agendas and suburbanisation is the loss of the peripheral forests of metropolitan centres. Literature on urban deforestation highlights the centrality of "ecosystem services" provided by urban forests from recycling to recreation, storage of nutrients, air filtering, and temperature regulation (Rusterholz et al. 2020). Additionally referred to as "forest landscape multifunctionality (FLM)", the qualities of urban forests guarantee water storage and biodiversity maintenance (Han et al. 2021).

Deforestation does not merely designate degradation of the ecosystem, it has detrimental effects on the livelihoods of forest communities, leading to struggle by communities observed worldwide over access to resources (Etchart 2022; Guha 2000; Hecht and Cockburn 2010; Rangan 2000; Schwartzman et al. 2010; Togami and Staggenborg 2022; Our History | The Green Belt Movement n.d.). Each case of resistance by forest villagers concerns a plethora of topics from overexploitation to environmental justice; from conservation to food security, cutting across divisions of gender, class, and ethnicity (Grasso and Giugni 2022). Not every resistance movement produces tangible outcomes. One of the immediate effects of deforestation is the extinction of key traditional occupations such as hunting, fishing, collecting wood, and gathering non-timber forest products, agriculture, aquaculture, and livestock, traditional medicine, food preparation, and crafts/skills (Status and

Trends in Traditional Occupations 2016). The loss of traditional forest knowledge (TFK) (Yinghe and Yeo-Chang 2021) further endangers the possibility of constructing a sustainable future based on global natural and cultural heritage (Berkes et al. 2002; Parrotta and Agnoletti 2007; Yuan et al. 2012; Farooquee et al. 2004).

In their work on the Potential effects of the EU's carbon border adjustment mechanism on the Turkish economy, Acar et al. suggest that if Turkey integrates macroeconomic policy that prioritises decarbonisation, she has the potential to increase national income while decreasing carbon emissions (Acar et al. 2022). Turkey's half-hearted concern with global climate change, however, is constrained by the logic of growth and the construction sector (Çelik 2021; Gülhan 2022; Karatepe 2021). Environmental damage to Istanbul's forests gains momentum with neoliberal policies, resulting in "mega projects" of rent generation such as the Third Bridge, the Istanbul Airport, Northern Marmara Highway, and Canal Istanbul, all occupying vast areas of northern Istanbul. These top-down projects not only play an instrumental role in the commodification of the state-owned lands, but they also give way to increasing socio-spatial problems, urban inequalities, and social segregation due to new luxury housing areas erected on privatised lands (Aysev 2022; Enlil and Dinçer 2022; Geniş 2007).

Loss of traditional occupations under urban transformation leads to commodification and proletarianisation of labour in former self-sustaining communities, resulting in the challenge of creating green and quality jobs that meet the standards listed in the CIPD Good Work Index (Good Work Index | Survey Reports n.d.). The complexity of the situation is further inherent in the degree of "greenness" of lost traditional work and the presumption that green industries can create sustainable jobs via government initiatives (Ernst et al. 2020; Sulich and Sołoducho-Pelc 2022; Green Jobs Initiative Report 2008; Woods et al. 2022).

The failure of top-down initiatives, such as the Navajo Green Jobs effort of 2009, suggests that socially and spatially embedded attempts at the green transition might lead to a better chance of success (Curley 2018). While the predominant criterion of green jobs relates to reduction in carbon emissions (Renner et al. 2008; Tănasie et al. 2022), questions remain as to whether green jobs are just, decent or safe, to what extent they can remedy social inequalities, and which skills will be required for the greening of the future work force (García Vaquero et al. 2021; Mulvaney 2014; Stilwell 2021; Strachan et al. 2023; Velicu and Barca 2020).

The present paper engages with this complexity by analysing work lost and created in a former forest community on the periphery of the mega city of Istanbul. We present this as a case study of the possibility of creating a just and green transition in local communities by grassroots initiatives even against the backdrop of extensive and aggressive deforestation and suburbanisation in a mega city fuelled by authoritarian developmentalism. We draw on studies on local initiatives reverberating 'quiet activism' (Pottinger 2016) that "prompted public awareness", such as the case of a local movement that instigated growing an indigenous wheat variety in the Aegean region of Turkey (Nizam and Yenal 2020). Viewing citizen participation as an integral part of sustainable transitions towards a circular economy (Melles et al. 2022; Velenturf and Purnell 2021), we underline the potential of such community projects to enhance place attachment and social wellbeing. The initiative under focus here, Community Fungi, can be seen as an example of the concept of "grounded knowledge" and participatory action research put forward by (Ashwood et al. 2014). The case also highlights the relevance of collective memories and place-making (Feola et al. 2023) to imagining a green and just transition. The following section describes the context of the study with a brief survey of the ecological and social transformation of the former forest village.

### 3. The Context

*In the Outskirts of the Megacity*

The case presented in this paper is a former forest village named Ömerli, occupying 4600 hectares of land, including 3800 hectares of the state forests in Northern Istanbul on the Asian side of the city (Figure 1).

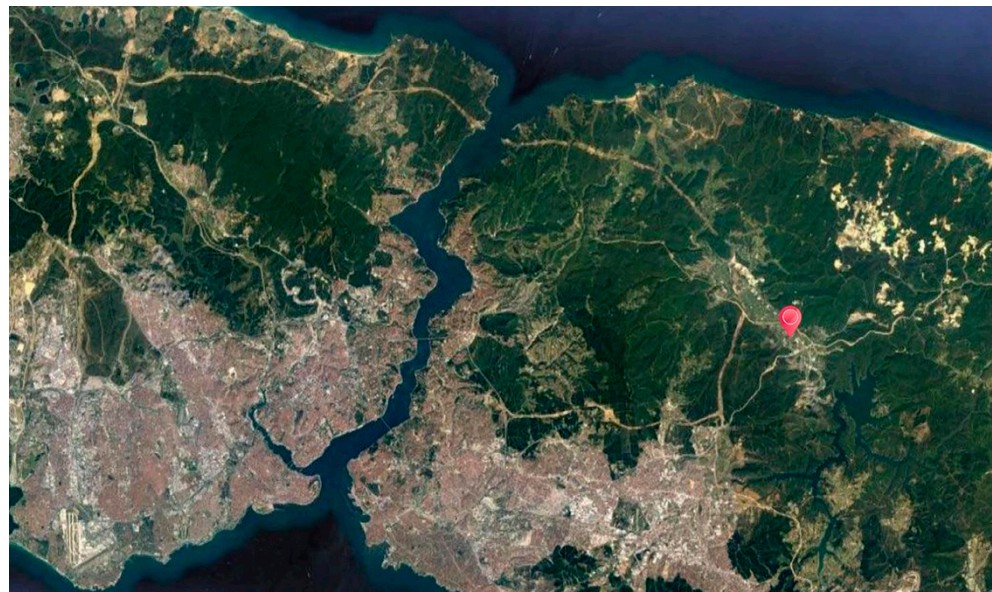

**Figure 1.** The location of Ömerli on the Asian side of Istanbul within the Northern Forests.

Ömerli formerly was home to a self-sustaining community of forestry, agriculture, and apiculture. This community experienced major changes from Ottoman "free" use of "the commons" (Hardin 1968) when the public enjoyed equal rights over the forest, to the "scientific forestry" of the late nineteenth century (Dursun 2007; Koç 2005) and deforestation of the Republican era. Central Istanbul underwent a profound construction program of major boulevards and coastal connection roads in tune with liberal policies implemented in the mid-20th Century as population growth and migration increased urbanisation (Gür and Dülgeroğlu Yüksel 2019). Meanwhile, settlements around the Northern Forests, including the Ömerli Basin, maintained self-sufficiency within a balanced ecosystem for at least three more decades. The urban transformation that followed, however, had detrimental effects on the regional forests, meadows, the watersheds, biodiversity, and the ecosystem.

Two discernible watersheds added momentum to this ecological and subsequent social transformation: the former is the construction of the Ömerli Reservoir in 1972, which dried up the river of the region; the second is the rise of gated communities with detached housing projects in the late 1990s and early 2000s, which began occupying agricultural and forest land. In tune with the rise of residential suburbs in Istanbul, the area suburbanised with proliferating middle-class detached and semi-detached housing that is distinct from the dominant residential configuration of multi-unit apartments of the city. Gated communities of detached housing in the region began with the "Kasaba Ömerli" project, the first phase of which was constructed during the late 1990s (Tanulku 2018). A variety of reasons including response to natural disasters such as the earthquakes of 1999 (Menteş and Töre 2020) and 2023, and the COVID-19 pandemic continue to precipitate demand for suburban housing and renders this sector lucrative. During the last two decades, over ten luxury detached housing sites emerged in the area (Figure 2).

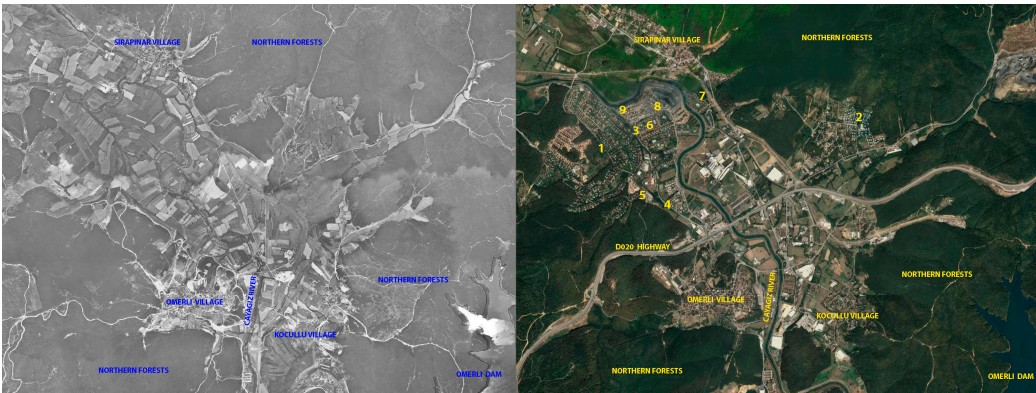

**Figure 2.** The map (from 1982) shows Ömerli and surrounding villages of Sırapınar and Koçullu within the Northern Forests of İstanbul, still undisturbed with many agricultural fields. The map, dating to 2022 shows Ömerli and surrounding villages within the Northern Forests of İstanbul, after the recent urban transformations.

The Northern Marmara highway and the connection roads to the Third Bridge prompted deforestation between 2012 and 2019, leading to the cutting of 3 million 700.000 trees. The oak coppice forests of Ömerli, along with the heathlands of the watershed, (Tezer et al. 2012), house an exceptional array of biological diversity for species, including animals, plants, and fungi. The Turkish Society for the Protection of Nature (DHKD) designates the region among Important Plant Areas (IPA) (Özhatay et al. 2005). Keeping the biodiversity of the northern ecological corridor is crucial for combating the effects of global warming (Aysev 2022), yet the natural habitat in the watershed was destroyed and water quality deteriorated (Tezer et al. 2012). The Northern Forests Research Centre documented over 300 environmental threats in the last three months of 2021, 69 of which were identified in Istanbul (Kuzey Ormanları Araştırma Derneği n.d.).

Along with ecological change, all local activities of self-sustenance in Ömerli retreated as the capitalist urban expansion of the city gained momentum from the mid-20th century, culminating in the rent-producing relationships and the authoritarian developmentalist agenda of the government in the last two decades. (Enlil and Dinçer 2022). This transformation resulted in proletarianisation of the former villagers who turned into service providers for middle-class gated communities.

As a significant difference between these two maps below, one dating to 1982 and the other to 2022 (Figure 2), the D020 Highway divided and disturbed the agricultural lands that once used to surround and feed Ömerli Village. A similar impact can be observed with the emergence of the detached housing projects, numbered from 1-9 on the current 2022 map, which are located between the Sırapınar Village and the D020 Highway. Industrial and mining sites are additionally visible on the 2022 map.

The present paper illustrates how the loss of vital ecosystem services due to deforestation transforms the lives of villagers by displacing forest workers. As small farms disappear and forests shrink, forest workers turn into commodified service providers. The question remains as to whether any space is left for old or new forms of work to be decent or green. More specifically, we explore possibilities grassroots micro-initiatives may offer towards generating decent and green jobs despite persistent threats to the environment.

## 4. Research Method

The present study is part of a larger ongoing project that was conceptualised as an extension of the Community Fungi (CF) initiative. This larger project aimed to examine the effects of urban transformation on Ömerli and its neighboring villages. In particular, the purpose was to account for the ecological effects of suburban development, mining, and industrial activities in the region as well as their perception by the inhabitants. The project began in December 2020 with fieldwork involving participant ethnography, continuing

visits to small local businesses and to homes, producing data based on ethnographic and oral history interviews that are being prepared for mixed methods analysis from an interdisciplinary perspective. The project has an action research element as one of the authors of the present paper is leading the interdisciplinary academic group and all authors are involved in it (Figure 3).

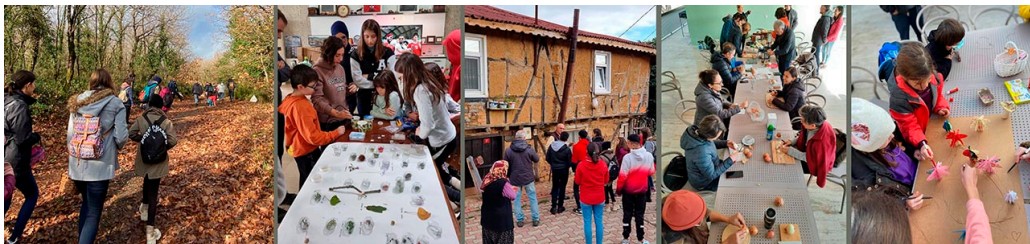

**Figure 3.** CF 2021, 2022 included science modules taking place during forest walks, collecting, and categorizing plant and mushroom samples and water analysis; cultural heritage modules involving neighbourhood walks, children making mycorrhizae and layered map models; and gastronomy and history modules involving collective cooking with local mushrooms.

The themes explored in the present paper emerged from this fieldwork as the investigation into the impact of ecological transformation on the community also generated insights about the extinct, surviving and newly created types of work in Ömerli. Building on previous participant observation, interaction, and semi-structured interviews, we conducted oral history interviews with four representative informants among the local community who gave consent, two men and two women. Each informant delivered their own narrative that reflects a certain type of experience within the ecological and social transformation of the village under scrutiny. The specific representative accounts analysed here provide some historical insights; however, we are aware of the clear partiality embedded in each one. These four interviews were carried out between 1 April–25 September 2022. The accounts of informants are subjected to qualitative analysis.

The male informants (Table 1 interviewee numbers 1 and 2), born within two years of each other, are both descendants of families with deeply rooted histories in the region since the Ottoman past. They are both eyewitnesses of the ecological and socio-economic transformation in question. They are professionals who chose to retire in the village, with pronounced sentiments of nostalgia. In this context, where written sources are rare, we adopt a critical realist approach to the accounts of interviewees 1 and 2 as they are our key informants, personal narratives of whom reflect parallel perspectives. They both represent middle-class privileged villagers who returned to their land with nostalgic views of their childhood and youth.

The female informants are from diverse backgrounds. Lale, (Table 1, interviewee number 3) joined the village community through marriage to Mehmet. Her agency in the community is noted as the educated facilitator and director of numerous certification courses that offer the training and skills necessary to produce and market handcrafts. The youngest among the interviewees (Table 1 interviewee number 4) is a working-class member of an ethnic minority and a representative of the proletarianisation of the villagers. Both interviewees 3 and 4 represent women's narratives and feminisation of labour in terms of necessary skills, limits, and possibilities for the remuneration of work performed by women.

We directed certain specific questions to the interviewees on the types of work they engaged in or witnessed in their lives. Alongside specific ones, we designed various open-ended questions to allow them to expand on their work and life experience. In addition to oral history interviews, we scrutinised relevant secondary literature, official documents on global, national, and regional ecological transformation, reports issued by non-governmental organisations, as well as testimonials and accounts voiced in different forms of media.

**Table 1.** Pseudonyms and short biographies of interviewees.

| Interviewee | Age | Gender | Pseudonym | Short Biography |
|---|---|---|---|---|
| 1 | 66 | Male | Mustafa | Local descendant of residents from one of the four major families among the earliest settlers of the Ottoman period. Amateur researcher of regional and family history; university graduate. Career: manager in a multinational corporate company. Current position: retired. |
| 2 | 64 | Male | Mehmet | Local descendant of residents from one of the four major families among the earliest settlers of the Ottoman period (not the same family as that of interviewee 1). Amateur researcher of regional and family history. Local administrator between 1998 and 2004, a family tradition since the late Ottoman era. University graduate. Career: chemist in corporate company; entrepreneur/owner of two local stores. Acted as neighbourhood administrator. Current position: retired. |
| 3 | 65 | Female | Lale | Moved to Istanbul for her secondary education from an Anatolian city in the late 1970s. Joined the Ömerli local community via marriage in the 1980s. University graduate; facilitator and director of local crafts courses offered to women. |
| 4 | 36 | Female | Burcu | Born and raised in Ömerli. Secondary school graduate. Female representative of new forms of work: cleaning and flower trade. |

## 5. Findings

### 5.1. The Abandoned Lifestyle and Extinct Forms of Work

Significant turning points regarding the extinction of some forms of work and the end of the past self-sufficient lifestyle emerge from the narratives of interviewees. These turning points coincide with the beginning of urban development with concrete architecture of multi-unit apartment buildings in central Istanbul in the 1950's; the end of free use of the forest commons in the 1970's; the construction of the reservoir in 1972; the connection of the village to the urban centre by roads; and the beginning of suburban development in the area from the 1990's onwards.

The urban development of central Istanbul had already begun before our oldest interviewees (Mustafa and Mehmet) were born in the late 1950s; however, they underlined the continuity of a self-sufficient lifestyle of the village throughout their childhood and youth in their testimonies. Both narratives portrayed a nostalgic view of their childhood in Ömerli where farming, animal husbandry, fishing, hunting, and free access to forest products sustained the self-sufficient community. They both claimed that the old lifestyle of the villagers was naturally eco-friendly, with centuries-old ways of caring for the environment.

From the perspective of Mehmet, in the mid to late 1960s, "Practically nothing was bought with money". In his recollection, the diet of the locals consisted generally of what was available, and if something was not locally grown, they simply did not eat it. Mehmet recalled commonly used stone ovens in the neighbourhood:

Following the communal bush gathering from the forest, women from each family brought their bread dough and collective baking took place once a week. Women also communally hand washed laundry in the fresh water from the hills, and harvested wheat and rice together.

Mustafa remembered similar scenes of female labour:

My grandmother used to weave linen cloth at home like other women of the village using a handloom. They hand dyed the textiles using oak leaves.

All three informants, Mehmet, Mustafa, and Lale, attested to the timber architecture of their family homes as "functional, convenient for communal domestic labour and climate friendly". Now under the threat of extinction, two-storey timber frame construction using

mudbrick as the wall-filling material on the facades dating back to the early 20th century was the natural choice for the forest villagers who had free access to the woods.

Mustafa and Mehmet mentioned that two significant sources of income were available to the local men during the period of free access to the forest. The first was the production and transportation of wood charcoal to the city, which was the only traditional income source. The carriages, the horses, and the carriage owners were supported by this business since the Ottoman era. Both informants narrated that before charcoal trade completely came to an end, it began to be replaced by the much more lucrative alternative of selling the local river sand to rising construction companies of Istanbul as of the 1950s. According to Mustafa, this business "seriously spoiled the villagers" who began to make unprecedented amounts of money because the river sand and gravel was especially favoured compared to sea sand, which constitutes construction hazards due to the salt content. Mustafa asserted that state ownership of the forests and replacing local forest villagers with officially appointed forest rangers put an end to this economic activity in the 1970s.

Along with these two major economic activities, Mustafa and Mehmet mentioned some other sources of income, the extinction of which, from their point of view, transformed social life in the village "for the worse". These included the open-air cinema and the local coffeehouse that used to serve "natural, healthy, home-made products". They recalled that the tradition of communal bread making had halted when a family from the Black Sea settled in Ömerli to establish the first bakery. Shoe repair was forgotten along with blacksmiths, who used to make horseshoes for carriage horses.

The following turning point was the construction of the Ömerli reservoir in 1972, which began to dry the local river's branch. The informants narrated that the villagers gradually abandoned river sand extraction because of this and the activity completely halted by the 1980's. The drying river due to the reservoir also caused willow trees to become extinct. Willow straw used to constitute a sturdy raw material for basket weaving by the women of the village. The informants put forward that the craft was abandoned, and the necessary skills were forgotten.

As the village was connected by highways and roads to the urban centre and suburban development of detached housing projects emerged in the 1990's, any remnants of subsistence economy and the past lifestyle disappeared. By the 2000's, the unleashed proliferation of concrete architecture led to the near extinction of craft skills and artisans (carpenters), who traditionally built and maintained the timber houses. Many timber buildings of Ömerli were in time dilapidated. The loss of traditional houses also marked the end of historically entrenched family labour and communal subsistence production, nostalgia for which came across the narratives of the informants.

The land plot of the earliest gated community of detached luxury housing, named Kasaba, was administratively a part of the Ömerli centre. The start of this project occupied a significant part in Mehmet's narrative, whose memories reflected resistance to hassled decisions aided by political actors with vested interests. From this perspective, Mehmet reported how the village land had started changing hands by an abruptly issued environmental impact assessment report (ÇED) that allowed the approval of local zoning plans and subsequent construction. Although construction land was not part of the forest, it was fertile arable land partly used for apiculture (Çevresel Etki Değerlendirmesi, İzin ve Denetim Genel Müdürlüğü n.d.).

> Our resistance to the environmental impact assessment report was not taken
> seriously. I made sure a critical commentary was added to the report, highlighting
> the arable quality of the construction site. First, they poured rubble on the land.
> Then they took samples from the soil and reported that the arable quality of the
> land was lost.

The loss of Ömerli's small-town status in 2006, to be annexed to the nearby Çekmeköy municipality, despite legal contestation, caused debilitating effects according to Mustafa:

This administrative change marked the beginning of the exploitation of Ömerli. It ended the practice of renting out the grazing land of the village pasture to provide income to maintain the community house and the Mosque. Construction activity that received approval from the municipality destroyed the grazing land.

All nostalgic accounts of the informants about the past described the bygone lifestyle as ecologically friendly and the forms of work from that period as decent and good. In very few instances, interviewees referred to non-decent elements of the past. No such interpretation emerged regarding the early, self-sufficient era from their youth. Extraction of local sand, which gained weight in the local economy only after Istanbul's urban development of concrete architecture, was criticized by Mustafa who described it as:

a strenuous activity that required bodily strength; and having observed many who fell ill, I suspect that it may have caused illnesses in workers born between 1945 and the 1960s.

While urban transformation and disappearing traditional ways of living are clear sources of concern from an ecological point of view, a less foreseen finding of this case study is that not every traditional work or communal practice that became extinct was green or decent. Our findings thus reveal a complex picture regarding extinct types of work and those that have been replacing the old "self-sustaining" lifestyle of the region. While charcoal trade provided an economically sustainable source of livelihood in the past, its environmental impact was not questioned by the informants. All interviewees viewed charcoal production as eco-friendly since forest villagers determined how many trees to cut, the choice of those trees and the volume of production based on traditional forest knowledge. Moreover, past communal practices romanticized for having ecologically friendly qualities integral to the "harmonious village community" indicate a significant dependence on unpaid female domestic labour. Mehmet's wife Lale reported that as late as the 1980's, when adverse effects of the water reservoir began to be felt, fresh water from the public fountain was a place to meet and chat with neighbours every morning. Lale's narrative reveals that unpaid domestic female labour, which included carrying water from the village fountain, continued to dominate women's lives well into the 1980's.

The following section looks at the present situation of new and surviving forms of work available for the residents of Ömerli.

### 5.2. The New and the Surviving Types of Work

5.2.1. Proletarianisation of the Villagers by Catering Services for Middle-Class New Residents

An immediately recognizable pattern in the present situation is the proletarianisation of local subsistence farmers, forest foragers, and traditional crafts people, along with the feminisation of labour with the proliferation of pink-collar jobs. Testimonies of our key informants suggest that this process gained momentum with the construction of detached housing projects and their gated communities.

Mehmet recalled that upon losing their land to construction companies, some villagers used their income from the sale of their property to invest in trucks, with the foreseeable prospect of leading teams of drivers to remove excavation dirt from the construction area. In his words:

For most of these investors, a feasible business idea led to insolvency due to the spiral of debt in which they fell. Having lost their property, many of them had to sell their trucks, ending up as wage labourers, as drivers of the trucks they previously owned. These makeshift drivers are today at the service of construction companies carrying rubble between the construction sites and dumping areas.

Mehmet and Mustafa reported that some poverty-stricken proletarianised locals and recent migrants from Eastern Turkey also sought jobs as construction workers in the development projects since the 1990s. We observe in the field that these jobs are shared with recent refugees, whose immigration status and lack of social security allow plunging wages.



While former apicultural, agricultural, and artisanal activities of the village ceased, changing men's part in economic activities, Lale's experience involved guiding village women towards vocational courses that earned them certificates in arts and crafts as the local director of the Istanbul Art and Occupational Training Courses (İSMEK n.d.) in 2006–2007. Lale recalled being an active facilitator of finding educators for numerous courses, including sewing and hairdressing classes. She recently revived basket weaving in the community centre.

A freshly rising economic activity of the area—clay excavation business—has become lucrative with increasing demand from European ceramic producers. Mining the clay requires considerable investment and equipment; in turn, remuneration is high, as global ceramic importers appreciate the clay quality of the region. Ceramic clay producers are alternative sources of employment, providing vocational training opportunities for workers. The ecological impact of such extractive industries that expand to the periphery of the village is further questionable but is not dealt with due to the limited focus of this study.

A more widespread phenomenon comprises jobs offered by gated communities to Ömerli's locals as security guards, cleaning personnel, nannies, makeshift caterers of food, and technicians to take care of repair work. A representative of the younger generation, Burcu described her childhood among the Roma community as a time of serious poverty. Today, just as her relatives and friends, she regularly works as a cleaning lady for these gated communities. Roma women's jobs as food caterers, cleaning personnel, and care takers are both precarious and relatively low-paid compared to those of immigrant workers (from the Philippines, Turkmenistan, Georgia, etc.) who are legally employed in the same households with compulsory social security and salaries paid in dollars. This picture is in tune with Sassen's treatment of transnational suburban service workers in global cities (Sassen 1988).

### 5.2.2. New Jobs Relying on Community-Based Resources

Apart from these more conventional pink-collar jobs, Burcu gave a detailed account of butcher's broom production, which recently became an area of specialisation for Roma families. Harvested from the local forest in November, the butcher's broom plant (cochina) is a valued product sold mostly to residents of the gated communities, especially around New Year's Eve, but also catered to greater Istanbul and exported abroad. Preparing the plant for sale is a labour-intensive task with multiple steps. Burcu revealed that:

> Harvesting cochina is hard work. It is only found in dense thornbush in the forest. We go in as a family and even if we wear gloves and boots, our entire bodies end up bleeding from the scars of the thorny plants. Immediately after harvest, my family enters a marathon of tying two different kinds of plants, the green bushes to the red flowers, aesthetically. This is difficult. Not everyone in the family is good at it. I am better, so I get to do a lot of the work. We continue to tie the flowers day and night, losing sleep for weeks throughout November.

This time-consuming activity also leads to the loss of precarious catering and cleaning jobs. After the product is ready, it is sold in bulk to wholesalers, who make a down payment to the families for approximately 10,000 cochina flower bunches per household. Nearly 100,000 bunches are sold to wholesalers for export every winter season. Burcu said that most Roma families were engaged in cochina production. Some families reserved part of the plants for themselves, and the women of these households became makeshift vendors of the plant during the winter holiday season, setting up their stalls at the entrances to the gated communities. Other marketing strategies involve using WhatsApp groups of gated communities. Burcu said that it was tough work, but it paid well. Family debt accumulated over a year was paid off within a month of cochina sales.

Picnic areas that provide barbeques and activities, such as horse riding for visitors and pony rides for children, proliferate in the area. These businesses also offer employment opportunities. In warmer weather, Burcu's husband works at a picnic area, guiding visitors through rides with his horse. He has a regular job as security personnel at a storage place,

and he continues to work as an on-call musician at picnic sites and wedding venues. Burcu adds that her brothers and her husband's friends also limit their shifts in their "day jobs," (as gardeners, drivers, or civil servants), leaving weekend and holiday schedules open for their musical performances. Considering Botazzi's analysis of contributive economy and justice (Bottazzi 2019), one way to raise Roma residents' well-being, autonomy, and work life quality is by playing music as a craft that they are raised with since their childhood.

### 5.2.3. The Surviving and the Revived Jobs

Despite profound environmental and socio-economic change, numerous types of traditional work from the early Republican period managed to survive, including the businesses of a men's and women's tailor, local restaurants, a bakery, a liquor store, numerous coffeehouses, a drugstore, three grocery stores, and small farm agriculture that allows villagers to attend the weekly farmers' market. The tailor's shop is a family business, the activities of which were almost reduced to repair and recycling. Driving trucks as service providers remains a source of income for a small number of men. The neighbourhood restaurant is run by the third generation of the same family. The cook/owner's son, despite viable alternatives, revived the restaurant that serves traditional recipes. Our informant Mehmet's son similarly left his corporate job to manage the family-owned neighbourhood store. These members of the younger generation of middle-class locals who return to the area are encouraged by their families who cling to nostalgia about the past. Their return to the old neighbourhood of the village to create a life or work environment away from the city, close to nature, is in tune with the rising movement of "new villagers" in Turkey motivated by sustainability and anti-consumption designs, migrating from urban centres to rural areas (Özdemir 2020). It is noteworthy that these educated young members of local families facilitate the survival and revival of old forms of work, rather than, in Richard Ocejo's words, continuing their involvement in "knowledge or technology-based 'good' jobs" (Ocejo 2017).

### 5.3. The Community Fungi Initiative

Continuous growth in detached housing projects attracts increasing numbers of new middle-class residents to Ömerli, with further adverse effects on the ecological balance. Given the additional misuse and misinterpretation of existing urban plans (Aysev 2022), we suggest that innovative forms of local participatory projects inspired by traditional sustainable practices may offer viable alternatives towards a path out of this dim picture. The Community Fungi initiative, comprising scholarly workshops first held in Ömerli in 2021, is such a case where, one of the co-authors of this paper is the organizer, and all three authors are facilitators. Community Fungi (CF) was conceptualised based on the experience provided by a precursor project led by the same scholar. This project, named Aegean Atelier (Ege'de Atölye https://egedeatolyedotorg.wordpress.com, accessed on 24 May 2023), was held between 2011 and 2018 in the Aegean region, around the central theme of the indigenous olive plant. The project gave rise to annual workshops in various olive-rich areas in the Aegean. These gatherings activated dialogue between scholars and local communities, a university course, and a popular science book (Blatchly et al. 2017) written by the project leader in collaboration with two liberal arts college scholars. The olive project also inspired the Olive Academy (Zeytin Okulu http://zeytinokulu.net, accessed on 24 May 2023) in 2016, a non-formal education platform and a physical space founded for promoting ecological literacy run by scholars and student volunteers.

Community Fungi workshops, named after the indigenous mushrooms of the Northern forests of Istanbul, comprise education activities to cultivate ecological literacy towards a sustainable future in the former forest village under scrutiny. Cutting across ethnic, gender, and class divisions, Community Fungi primarily aims to bring together local villagers, residents of gated communities, and scholars, highlighting vital ecosystem services by inspiring the co-creation of site-specific decent and green jobs.

In this context, the Community Fungi initiative assumes three major functions: The first is to provide material knowledge about the environment based on traditional forest knowledge (TFK), including familiarity with water sources, indigenous plants and animals, and corresponding recycling methods, housing, gardening, and cooking techniques. The second is to promote a collective imagination of a green transition for the local residents inspired by traditional communal practices aiming to perpetuate vital forest ecosystem services (Rusterholz et al. 2020). The third is to lay the groundwork to inspire and support the co-creation of decent and green jobs and skills. To this end, rising middle-class gated communities with their consumer demands can lead to plans to create various green and decent jobs, including but not limited to ecological cleaning, gardening, catering, and providing local knowledge of an eco-friendly lifestyle.

More specifically, with the prospect of fostering an environment of ecological literacy and dialogue, CF 2021 aimed at nurturing communication and collaboration among invited scholars, local residents, and members of gated communities around the theme of mushroom varieties of the Northern Forests. It took place in a gated community social centre and the surrounding forest. The fungi theme was the choice for the first festivity, as mushrooms are ubiquitous and integral parts of the vital ecosystem services. They enable natural decay and recycling of life in the forest, while being gastronomically appealing to local foragers as well as ecotourism entrepreneurs (Tsing 2015). Mycorrhiza was the central concept in 2021, as it unites the forest under one large cooperating network of tree roots and fungi. A session on Anna Tsing's *The Mushroom at the End of the World* moved the discussion towards the ecological destruction of the earth and Ömerli's place in the big picture. Water emerged as a future topic, as the Ömerli reservoir provides one third of Istanbul's clean water. The workshop raised awareness among participants about the correlation between increasing population and water contamination. CF discussions led to the accumulation of local material knowledge on the environment, triggering interdisciplinary projects, such as the current study.

Community Fungi 2022 was held in the village centre with the participation of local residents and members of surrounding gated communities. Discussions and networking took place towards promoting a collective imagination of a green transition for both the local and new residents inspired by communal practices and traditional forest knowledge (TFK). The concurrent discussion on water led to queries on cleanliness, contamination, and irrigation, and future meetings are being considered on ecological literacy, skills, and entrepreneurship opportunities for cleaning personnel and homeowners residing in Ömerli with the intent of creating green jobs. A parallel development concerns catering businesses of local women, focusing on indigenous plant, mushroom, herb, and cheese varieties. Part of this discussion concerns creating cooperative ways of matching demand from middle-class residents and working-class caterers to reduce the precariousness of their business. Another dimension involves recording and documenting local knowledge of traditional recipes, in collaboration with a local eco-tourism entrepreneur and scholars of food history.

## 6. Conclusions

The globally observed authoritarian developmentalism that fosters neoliberal construction projects continues to expand urbanisation and suburbanisation of mega cities. Within this general picture, a green transition, or creating green economies, is hardly prioritised. Turkish economic programs of the last two decades based on the commodification of state lands offer no exceptional perspective to the global picture (Arsel et al. 2021).

The case study of a former forest village in the periphery of the city of Istanbul featured in this paper shows that ecological transformation, suburbanisation and middle-class migration placed pressures on the villagers to turn into wage labourers in precarious service and pink-collar jobs, feminising labour. Decent, green jobs are far from available for the working-class residents of the former forest community.

The present study is part of a larger project that continues to assess the ecological and social effects of suburban development, as well as extractive and industrial activities in

the region. A major limitation of the present paper is due to restricting the analysis to the impact of urban development on the types of work available to the village community. Industrial plants and mining activities in the periphery of the village are not accounted for in this study. This limitation gives an incentive for future research that will combine the findings from the present paper with the larger project for an in-depth look at an extensive area.

Despite the thin prospect of macroeconomic and political reversal toward providing green and decent work alternatives in the short run, we suggest that a just transition to this end may be triggered by participatory grassroots micro-initiatives among villagers and new residents. Community Fungi, as a grassroots initiative of the village we examined, illustrates the prospect of a collective imagination towards a green transformation with green and decent jobs.

Finally, Community Fungi holds the potential to open new venues of communication and collaboration with similar urban transformation cases and regional, national, and global grassroots projects to foster the collective imagination of a green future.

**Author Contributions:** Conceptualization, İ.S., B.S.C.; methodology, İ.S.; validation, İ.S., B.S.C. and Z.D.N.; investigation, İ.S., B.S.C. and Z.D.N.; resources, İ.S., B.S.C. and Z.D.N.; data curation, İ.S., B.S.C. and Z.D.N.; formal analysis, İ.S., Z.D.N. and B.S.C.; writing—original draft preparation, İ.S., B.S.C. and Z.D.N.; writing—review and editing, İ.S., B.S.C. and Z.D.N.; visualization, B.S.C. and Z.D.N.; project administration, İ.S. and Z.D.N. All authors have read and agreed to the published version of the manuscript.

**Funding:** This research received no external funding.

**Institutional Review Board Statement:** The study was conducted in accordance with the Declaration of Helsinki and approved by Özyeğin University Research Ethics Committee (13 April 2022).

**Informed Consent Statement:** Written informed consent was obtained from all subjects involved in the study.

**Data Availability Statement:** Not applicable.

**Acknowledgments:** The authors thank Berna Zengin Arslan for her valuable comments on the manuscript. They are also grateful to the oral history interviewees for their generosity with their time throughout the interviews.

**Conflicts of Interest:** The authors declare no conflict of interest.

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
