# Peer review of "Imagining Decent Work towards a Green Future in a Former Forest Village of the City of Istanbul"

_socsci, doi:10.3390/socsci12060342_

Round 1

Reviewer 1 Report

This paper presents a case study on the on-going ecological deterioration in a forest community in Turkey and the potential of collective action in sustaining rural livelihoods and decent works in this transformation process. The study is based on an emprical research conducted in Ömerli Village, İstanbul. Data drawn from the in-depth interviews with local residents in this village.

The authors rightfully acknowledges a connection between the reaction among locals to changing conditions and gendered/racialized hierarchies in the local community. The authors’ broader goal is to present a critical analysis of existing hierarchies in the local community and their impact on the new possibilities of decent work for the livelihoods of local communities. There are some very important ideas and rich ethnographic account in this paper that warrant publication. What I would like to advise is to strengthen the claims about the significance of the arguments, in other words the author may think to develop their arguments in a way to show the universal significance of the arguments!

The below are some minor issues:

1) It is not clear what the authors means by the concept of resistance. Whether they refer to a resistance against ecological deterioration or a rapid social change? Or whether they mean “resilience”?

“…we suggest that as certain types of work are abandoned and new forms are created, adaptation to the ecological and social conditions prevent resistance” (page 1) .

“While reasons for the lack of organized resistance to ecological deterioration are briefly considered here, focusing on the forest village under scrutiny we suggest that adaptation to the new circumstances by abandoning old types of work and creation of new ones gendered and ethnic hierarchical forms eclipse the potential of resistance”. (1-2)

2) To maintain confidentiality the author should avoid giving the details of the identity of the research participants, e.g. their family history.

3) The authors focus on “the Community Fungi” as a community-building project. The community development or community building is still very novel issue/or concept in Turkey. It would be good that the authors explain this concept or add a few words about the importance of this approach/policy for social change.

Reviewer 2 Report

This paper is a decent contribution to the debate on the sustainability of different forms of work in the case of the ecological and social transformation. It sheds light on a highly important issue in a country in Global South, Turkey where ecological deterioration is highly severe and the social transformation related to this deterioration has caused changes in the survival strategies of the local community. I definitely enjoyed reading it and recommend the paper to be published after a revision. 

Some concrete remarks:

1. The main problem of the paper in my opinion is, it lacks a clear research objective. It starts with a statement to argue the future of work and the different types of work which survived or abandoned in the forest village after the ecological and social change which should be the main objective as this is supported with the data later in the paper. However, the authors try to discuss another important issue inadequately here and there which is the lack of the resistance movements that appear against the ecological change. Especially in the introduction we are not sure about the main purpose of the paper which should be clarified. A sentence which explains what is following in the next sections would be beneficial for a better flow. The 'Ecological Change' section starts with the discussion on different resistance practices experienced in the world but the reasons why there is not an effective movement in Omerli is not identified sufficiently. Therefore  the resistance issue should either be strengthened or mentioned only as a dimension of the issue not as a main argument of the paper. 

2. The number of the interviews is rather low for the research although the authors defined the study as part of a large research. It could have been more suitable methodologically, if the data collection method was identified as 'oral history' interviews rather than semi-structured interviews and giving the duration of the interviews would be better to justify the low number of interviews. 

3. I have a problem here with the use of 'communal labour'. As the paper lacks a proper theoretical framework (which is a crucial point), as readers we are not satisfied with the definitions -or lack of definitions in this case- of 'labour' and 'work'. As it is exemplified here hand-washing laundry or baking bread are hardly communal labour but communal activities as part of the abandoned village lifestyle. If they used the Marxist definition of reproductive female labour, there might have been a room for a theoretical discussion here. But the authors have not explained the different types of labour theoretically. 

4. In the discussion section we see the resistance debate again although we have hardly read any explanation in the findings. In general the discussion introduces really important debates which we haven't seen anywhere in the paper before such as the feminisation of labour among Roma women. 

5. Overall it would be beneficial if the authors make a good connection between the sections and clarify their arguments and focus on the aim/aims throughout the paper. 

Round 2

Reviewer 2 Report

The paper has been significantly improved. It has a clear argument and focus in addition to a balanced discussion now. I am looking forward to seeing the published version.